# Identification of Linear B Cell Epitopes on CD2V Protein of African Swine Fever Virus by Monoclonal Antibodies

Rui Jia,[a] Gaiping Zhang,[a,d,e] Yilin Bai,[c] Hongliang Liu,[a,b] Yumei Chen,[a] Peiyang Ding,[a,b] Jingming Zhou,[a] Hua Feng,[b] Mingyang Li,[b] Yuanyuan Tian,[a] Aiping Wang[a,b]

[a]School of Life Sciences, Zhengzhou University, Zhengzhou, Henan, China
[b]Henan Zhongze Biological Engineering Co. Ltd., Zhengzhou, Henan, China
[c]Northwest Agriculture Forestry University, Yanglin, Shanxi, China
[d]Peking University, Beijing, China
[e]Longhu Modern Immunity Labrotary, Zhengzhou, Henan, China

**ABSTRACT** The CD2-like (CD2V) protein is a crucial antigen of African swine fever virus (ASFV). CD2V interacts with the cellular AP-1 protein, participates in intracellular transport of virus, and induces neutralizing antibodies to partly protect swine from virus attack. In this study, a specific CD2V dimeric protein was designed to enhance antigenicity and immunogenicity, expressed in a Bac-to-Bac baculovirus expression vector system and purified by Ni-affinity chromatography. After animal immunization, five monoclonal antibodies (mAbs) (7E12, 22B3, 18A3, 13G11, and 43C2) against CD2V were developed. The variable regions of heavy chains and light chains of the mAbs were sequenced to prove that the five mAbs differed from one another. The mAbs of CD2V could combine with ASFV by immunoperoxidase monolayer assay (IPMA). B cell epitopes of CD2V were screened using the five mAbs by indirect enzyme-linked immunosorbent assay (ELISA) and Dot-ELISA. Therefore, three B cell epitopes ([147]FVKYT[151], [157]EYNWN[161], and [195]SSNY[198]) were identified. This is the first time that mAbs of the ASFV CD2V protein have been developed and the sequencing of heavy chains and light chains of mAbs has been completed. Linear B cell epitopes, which were core targets of immunoprotection of the CD2V protein, were identified by mAbs for the first time. This study provides efficient epitopes for the development of ASFV subunit vaccines.

**IMPORTANCE** The ASFV CD2V protein is a crucial antigen on the outer envelopes of virus particles. A modified ASFV CD2V dimeric protein was expressed in the Bac-to-Bac baculovirus expression vector system. Five monoclonal antibodies (mAbs) against CD2V were developed, sequenced, and applied to identify CD2V protein B cell epitopes. Three B cell epitopes, [147]FVKYT[151], [157]EYNWN[161], and [195]SSNY[198], were identified. This is the first time CD2V mAbs have been developed, the sequencing of heavy chains and light chains of CD2V mAbs have been completed, and CD2V B cell epitopes have been identified by using scanning peptide method and bioinformatics methods.

**KEYWORDS** ASFV, CD2V, monoclonal antibodies, B cell epitopes

African swine fever (ASF) is a highly pathogenic, lethal, and contagious viral disease caused by African swine fever virus (ASFV) (1, 2). ASF can infect domestic pigs and wild boars of any breed and age and cause high fever, bleeding of the reticuloendothelial system, and even death (3). It was first reported that a wild boar (*Sus scrofa domesticus*) was infected by ASF in Kenya in 1921 (4, 5). Subsequently, ASF spread quickly to other countries in Europe (6–8) and Asia (9–12). In 2018, ASF was introduced to China, which was the largest country contributing to the swine industry. That brought huge losses to the pig industry (13). The swine industry was related to the healthy development of global food security (14).

Address correspondence to Gaiping Zhang, zhanggaip@126.com, or Aiping Wang, pingaw@126.com.

The authors declare no conflict of interest.

*[This article was published on 21 March 2022 with affiliations d and e missing for Gaiping Zhang. The affiliations were updated in the current version, posted on 28 March 2022.]*

ASFV, the pathogenic agent of ASF, is the only member of the *Asfarviridae* family and the *Asfivirus* genus (15). ASFV is a linear, covalently closed double-stranded DNA virus with a genome size varying from 170 kbp to 190 kbp (15–18). The structure of the ASFV virion is a symmetrical icosahedron with a diameter of approximately 200 nm. ASFV is also a large nucleocytoplasmic DNA virus (NCLDV) (2, 18). The genome of ASFV has hundreds of open reading frames (ORFs). ASFV is a complex multienveloped virus encoding 151 to 167 proteins. These proteins include more than 50 structural proteins, which are located on different envelopes and involved in genome replication and viral infection (4, 19).

The ASFV CD2V protein, which is encoded by the EP402R gene, is a glycosylated protein that plays an important role in viral pathogenesis (20), orientation of the host (21), and immune escape (22). CD2V protein is an important protective antigen of ASFV, which provides ASF with serotype-specific cross-protective immunity (23). Immune pigs with ASFV CD2V protein can produce HAI and antibodies against M-II, which can be partly protected during challenge with homologous virus strains (24, 25). Most immunogenicity is determined by epitopes, which are the main chemical substances recognized by the immune system. Epitopes recognized by B cells, T cells, and soluble antibodies are the core of the immune response. B cell epitopes refer to the regions of protein molecules recognized by antibodies. T cell epitopes are short peptides recognized by T cells after binding to MHC molecules. The immunoprotective effect seems to be related to the titer of M-II antibody (26). At present, T cell epitopes on the CD2V protein have been identified (27, 28). However, there has been no research on the B cell epitopes of the CD2V protein.

In this study, we designed a specific CD2V recombinant protein to improve antigenicity and immunogenicity. The CD2V recombinant protein was expressed in the Bac-to-Bac baculovirus expression vector system and purified by Ni-affinity chromatography. After animal immunization, monoclonal antibodies against CD2V were developed to screen B cell epitopes of CD2V. There is currently no effective and safe vaccine against ASFV (2). The preparation of antibodies and the identification of B cell epitopes provide theoretical support for the development of subunit vaccines and diagnostic reagents.

## RESULTS

**Expression and purification of CD2V recombinant protein.** The results on bioinformatics analysis and codon optimization of the CD2V recombinant protein are shown in Fig. S1 and S2 (supplemental material). The result on constructing recombinant vector pFastBac 1-CD2V is shown in Fig. S3 (supplemental material). Expression of CD2V recombinant protein was detected in the cell supernatant of third-generation baculoviruses. The size of the CD2V recombinant protein was identified to be approximately 46 kDa by SDS-PAGE (Fig. 1A) and Western blotting (Fig. 1B). Indirect immunofluorescence assay (IFA) indicated that CD2V recombinant protein was expressed well (Fig. 1C). As for the purification of CD2V, washing buffer containing 10 mM imidazole was used to wash away the impurities, and the target protein was collected with elution buffer containing 50 mM imidazole (Fig. 1D). Western blotting (Fig. 1E) showed that the eluted protein is the target protein. Dot-ELISA (Fig. 1F) also showed that different doses of CD2V protein could react with positive serum from rehabilitated pigs with ASFV. Size exclusion chromatograph showed that the molecular weight of CD2V recombinant protein was about 46 kDa according to the molecular weight of Ferritin ($M_r$ 440,000) among seven model proteins (Fig. 1G). CD2V recombinant protein particles aggregated into approximately 10-nm nanoparticles (Fig. 1H) which were observed by 120 kV transmission electron microscopy (TEM). As is shown in Fig. 1I, the particle size of CD2V recombinant protein was 10 nm. It was shown that CD2V had formed nanoparticles in solution, which was likely caused by posttranslational modification. As was shown in Fig. 1J, the reaction titer of the CD2V protein with the positive serum was 1:6,400 by indirect ELISA. It was also shown that CD2V protein could react with ASFV-positive serum. The results of the immunogenicity analysis of the CD2V

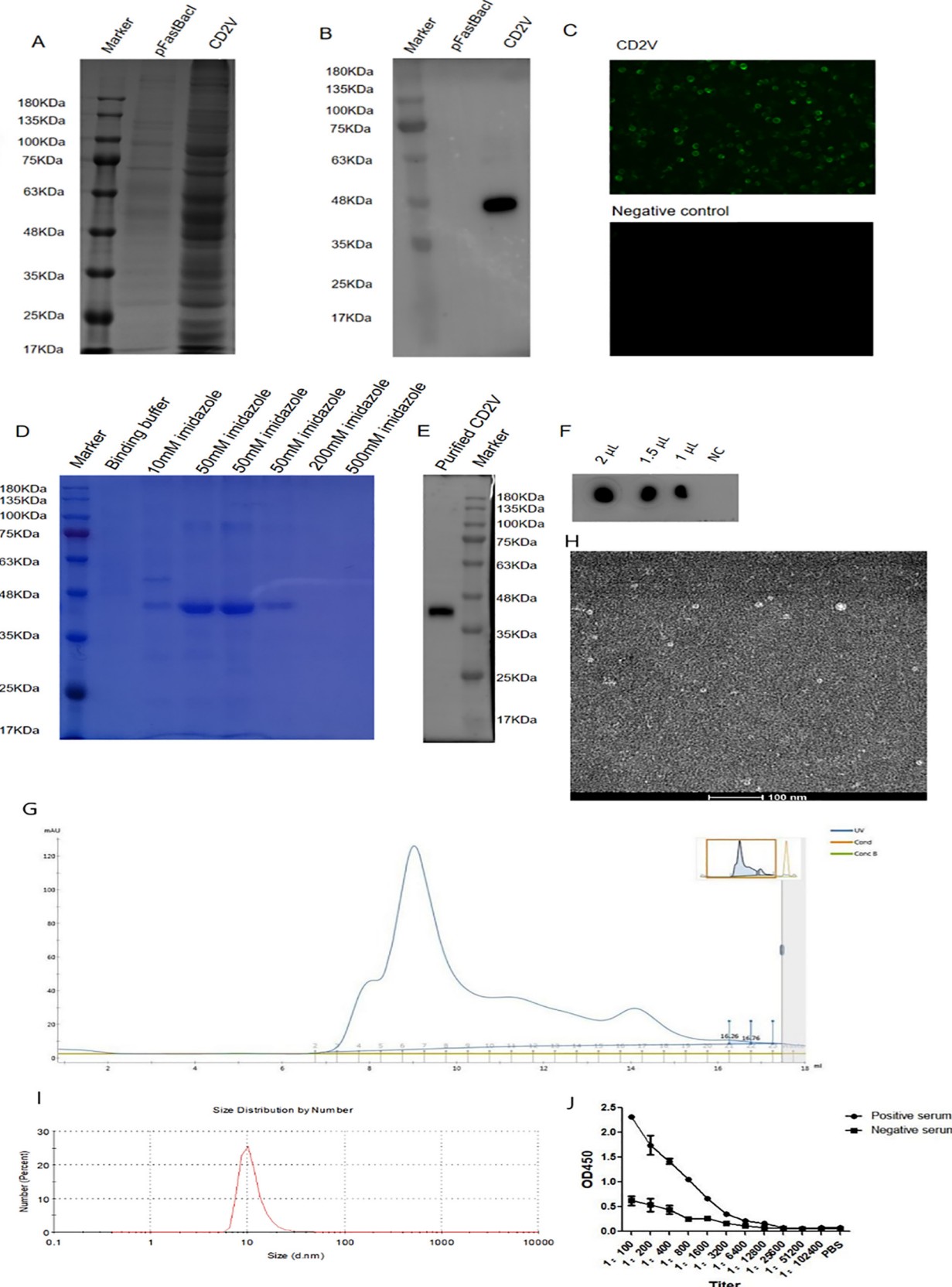

**FIG 1** Expression and purification of the CD2V recombinant protein in the Bac-to-Bac baculovirus system. Results of SDS-PAGE (A), Western blotting (B), and IFA (C) indicated that CD2V recombinant protein was expressed well in the Bac-to-Bac baculovirus system. Results of SDS-PAGE

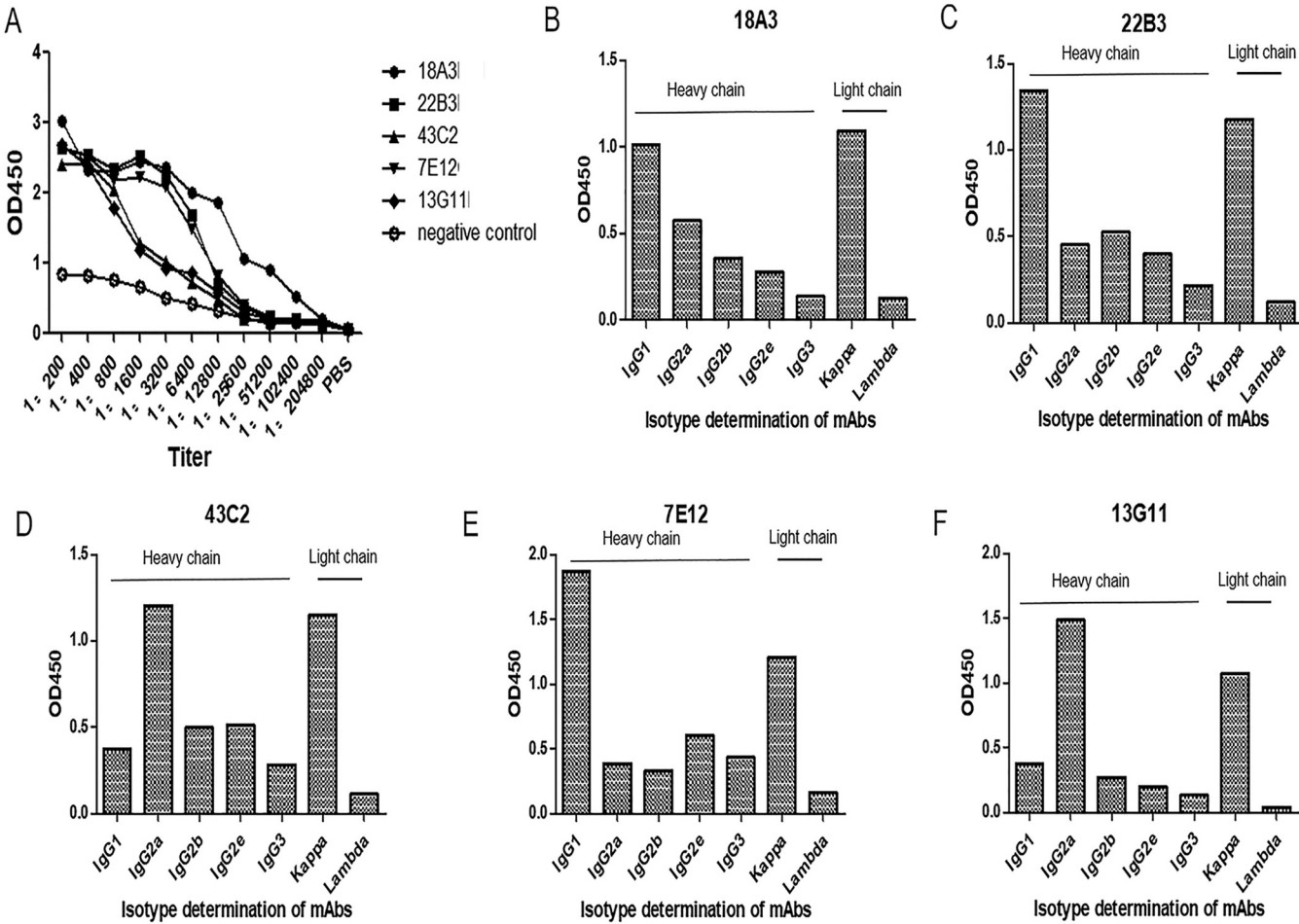

**FIG 2** Titers and isotype identification of mAbs. (A) Titers of five mAbs, 18A3, 22B3, 43C2, 7E12, and 13G11, were 1:1.024 × 10$^6$, 1:1.28 × 10$^5$, 1:6.4 × 10$^4$, 1:1.28 × 10$^5$, and 1:6.4 × 10$^4$, respectively. (B) The heavy- and light-chain isotypes of 18A3 were IgG1 (OD$_{450}$ of 1.016) and kappa (OD$_{450}$ of 1.094). (C) The heavy- and light-chain isotypes of 22B3 were IgG1 (OD$_{450}$ of 1.343) and kappa (OD$_{450}$ of 1.179). (D) The heavy- and light-chain isotypes of 43C2 were IgG1 (OD$_{450}$ of 1.206) and kappa (OD$_{450}$ of 1.151). (E) The heavy- and light-chain isotypes of 7E12 were IgG1 (OD$_{450}$ of 1.869) and kappa (OD$_{450}$ of 1.209). (F) The heavy- and light-chain isotypes of 13G11 were IgG1 (OD$_{450}$ of 1.489) and kappa (OD$_{450}$ of 1.070).

recombinant protein are shown in Fig. S4 (supplemental material). Hence, our results indicated that CD2V recombinant protein expressed in Bac-to-Bac baculovirus expression vector system had strong immunogenicity and immunoreactivity.

**Screening and identification of monoclonal antibodies against CD2V recombinant protein.** Five monoclonal antibody cells with high titers were screened by indirect ELISA as shown in Fig. 2A and Table 1. Titers of 18A3, 22B3, 43C2, 7E12, and 13G11 were, respectively, 1:1.024 × 10$^6$, 1:1.28 × 10$^5$, 1:6.4 × 10$^4$, 1:1.28 × 10$^5$, and 1:6.4 × 10$^4$. The heavy-chain isotypes and light-chain isotypes of 18A3 (Fig. 2B) were IgG1 (optical density at 450 nm [OD$_{450}$] of 1.016) and kappa (OD$_{450}$ of 1.094). The heavy-chain isotypes and light-chain isotypes of 22B3 (Fig. 2C) were IgG1 (OD$_{450}$ of 1.343) and kappa (OD$_{450}$ of 1.179), and the heavy-chain isotypes and light-chain isotypes of 7E12 (Fig. 2E) were IgG1 (OD$_{450}$ of 1.869) and kappa (OD$_{450}$ of 1.209). The heavy-chain isotypes and light-chain isotypes of 43C2 (Fig. 2D) were IgG1 (OD$_{450}$ of 1.206) and kappa (OD$_{450}$ of 1.151), the heavy-chain isotypes and light-chain isotypes of 13G11 (Fig. 2F) were IgG1 (OD$_{450}$ of 1.489) and kappa (OD$_{450}$ of 1.070).

**FIG 1** Legend (Continued)

(D) and Western blotting (E) for purification of the CD2V recombinant protein, respectively, showed that CD2V recombinant protein was purified by Ni-affinity chromatography. (F) Dot-ELISA for CD2V recombinant protein purification. (G) The molecular weight of CD2V protein was 46 kDa by size exclusion chromatography. (H) The CD2V protein was collected to 10 nm nanoparticles by 120 kV TEM. (I) Particle size of CD2V was approximately 10 nm in solution by Zeta CAD. (J) The reaction titer of the CD2V protein with the positive serum was 1:6,400 by indirect ELISA.

**TABLE 1** Identification for subtypes of monoclonal antibodies[a]

| mAb | Heavy-chain subtype | Titer |
|---|---|---|
| 7E12 | IgG1 | $1:6.4 \times 10^4$ |
| 22B3 | IgG1 | $1:1.28 \times 10^5$ |
| 13G11 | IgG2a | $1:1.28 \times 10^5$ |
| 18A3 | IgG1 | $1:1.024 \times 10^6$ |
| 43C2 | IgG2a | $1:6.4 \times 10^4$ |

[a]All mAbs had the light-chain subtype.

Western blotting and IFA are shown in Fig. 3. All five mAbs reacted with the CD2V protein, and specific bands were at approximately 46 kDa. Among them, 22B3 (Fig. 3C), 13G11 (Fig. 3E), and 18A3 (Fig. 3G) reacted strongly with the CD2V protein, while 7E12 (Fig. 3A) and 43C2 (Fig. 3I) reacted weakly. The reaction between ASFV-positive serum and CD2V protein is shown in Fig. 3K. Each band on Western blots of CD2V recombinant protein with the monoclonal antibody was wider than that of CD2V recombinant protein with ASFV-positive serum, which showed that CD2V recombinant protein had the better reactivity with the monoclonal antibody. All the mAbs, including 7E12 (Fig. 3B), 22B3 (Fig. 3D), 13G11 (Fig. 3F), 18A3 (Fig. 3H), and 43C2 (Fig. 3J), reacted strongly with the CD2V protein which was expressed in sf21 cells.

**Sequence identification of monoclonal antibodies.** Nucid sequences of heavy-chain and light-chain variable regions were amplified by PCR (Fig. S5). As is shown in Table 2, the sequences of the five CD2V recombinant protein mAbs were different from each other. Sequences of CDRs on heavy-chain and light-chain variable regions of mAbs were verified in IMGT online software (Table 2). The results of IMGT analysis are shown in Fig. S6.

**Titration of mAb staining of infected monolayers by IPMA.** Immunoperoxidase monolayer assay (IPMA) neutralization assay is shown in Fig. 4. CD2V mAb 18A3 (Fig. 4A), 22B3 (Fig. 4B), 43C2 (Fig. 4C), 7E12 (Fig. 4D), and 13G11 (Fig. 4E) as well as ASFV-positive serum (Fig. 4F) could bind to ASFV HLJ/18 strain. However, pig negative serum (negative control) and phosphate-buffered saline (PBS; blank control) could not bind to ASFV HLJ/18 strain. As is shown in Fig. 4A and B, IPMA titers of 18A3 and 22B3 were both 1:8,000. As is shown in Fig. 4C and D, IPMA titers of 43C2 and 7E12 were 1:4,000. As is shown in Fig. 4E, IPMA titer of 13G11 was 1:2,000. As is shown in Fig. 4F and G, IPMA titer of positive recovery serum was 1:8,000, while negative serum and PBS did not react with virus.

**Preliminary identification of CD2V protein B cell epitopes.** The results of bioinformatics prediction for B cell epitopes of CD2V are shown in the supplemental material and Fig. S6. The reactions of anti-CD2V mAbs with 18 short peptides in three repeated tests are shown in Fig. 5. The no. 14 short peptide ([147]FVKYTNESILEYNWN[161]) reacted with 7E12 (Fig. 5A), 22B3 (Fig. 5B), 13G11 (Fig. 5C), 18A3 (Fig. 5D), and 43C2 (Fig. 5E), and the $OD_{450}$ values of these reactions were above 1.0. No. 18 ([187]INCTYLTLSSNYTFFKLY[206]) had stronger reactions with 7E12 (Fig. 5A), 22B3 (Fig. 5B), 18A3 (Fig. 5D), and 43C2 (Fig. 5E) than with 13G11 (Fig. 5C), and the $OD_{450}$ values were approximately 1.0. As is shown in Fig. 5F, the negative mAb WH303 did not react with peptides and CD2V recombinant protein (all the $OD_{450}$ values were lower than 0.5). In Fig. 5, the error bars refer to standard deviation (SD) of three replicates from one experiment. As is shown in Fig. 5G to K, dot-ELISA was used to verify the above results. Membranes were dotted by peptides 1 to 18. The results showed that 7E12 (Fig. 5G), 22B3 (Fig. 5H), 13G11 (Fig. 5I), 18A3 (Fig. 5J), and 43C2 (Fig. 5K) reacted well with no.14 peptide and positive serum, while 7E12, 22B3, 18A3, and 43C2 reacted well with no. 18 peptide. These results were consistent with the results of indirect ELISA (i-ELISA) identification. These findings indicated that no.14 short peptide ([147]FVKYTNESILEYNWN[161]) and no. 18 short peptide ([187]INCTYLTLSSNYTFFKLY[206]) were B cell epitopes of ASFV CD2V protein.

**Truncation of short peptides.** As shown in Fig. 6A and B, the secondary structures of no. 14 peptide and no. 18 peptide were predicted by DNASTAR software. No. 14 peptide was truncated into three peptides, 14-1 ([147]FVKYT[151]), 14-2 ([151]NESIL[156]), and 14-3 ([157]EYNWN[161]), and no. 18 peptide was also truncated into three segments, 18-1

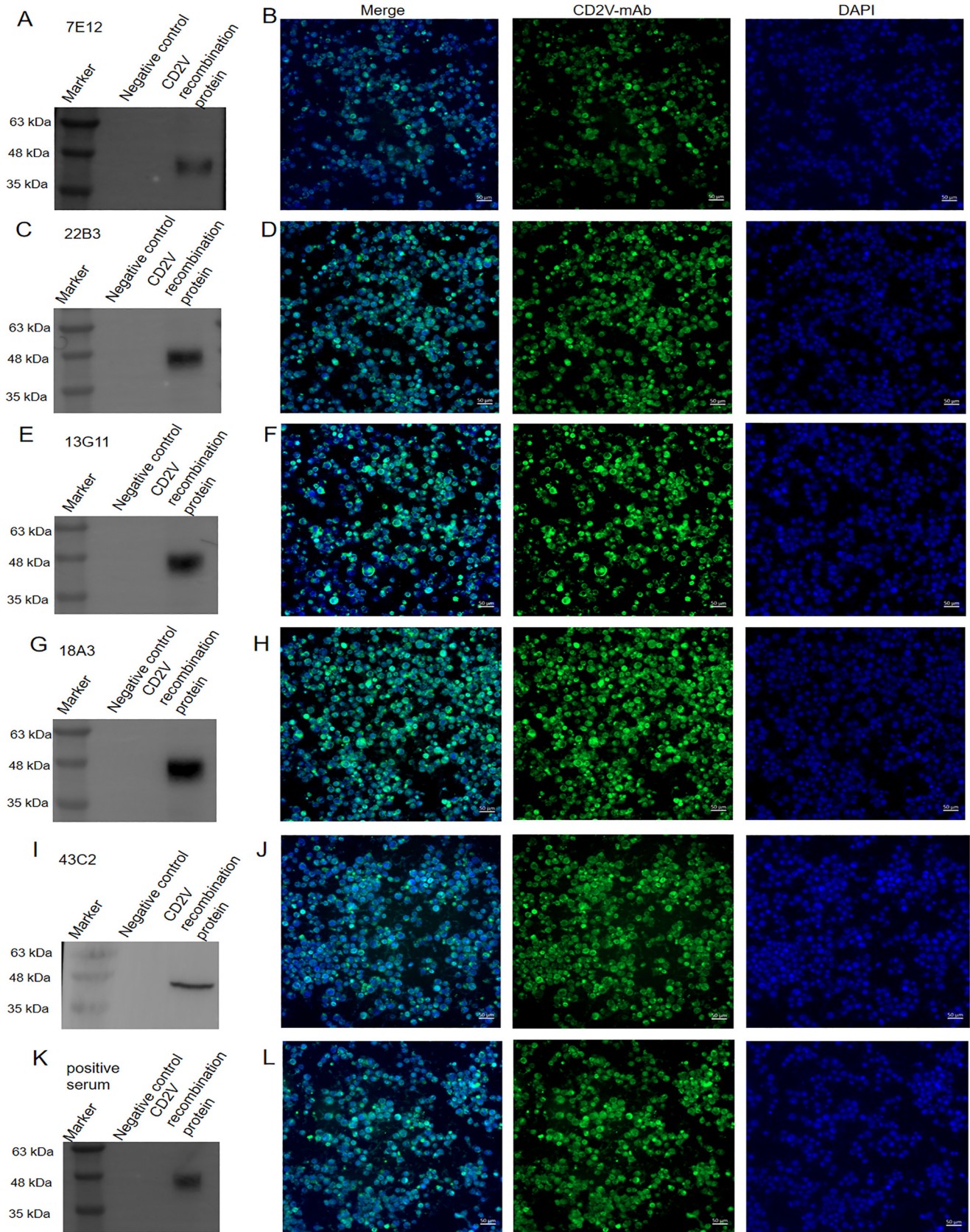

**FIG 3** Detection of the reaction between the mAbs and the CD2V recombinant protein using Western blotting and IFA. The 7E12 mAb could react with the CD2V recombinant protein by Western blotting (A) and IFA (B). The 22B3 mAb could react with the CD2V recombinant protein by Western blotting (C)

**TABLE 2** Variable region of heavy chain and light chain of CD2V mAbs

| mAb | Variable region of heavy chain | Variable region of light chain |
|---|---|---|
| 18A3 | MVKPGASVKISCKASGYSFTSYYIHWVKQRPGQGLEWIGWIFP GSGNTKYNEKFKGKATLTADTSSSTAYMQLSSLTSEDSAVYFC AQTGRVFAYWGQGTTVTVSSK<br>CDR-H1:GYSFTSYY<br>CDR-H2:IFPGSGNT<br>CDR-H3:AQTGRVFAY | MTQSPSSLSASLGERVSLTCRASQEISGYLSWL QQKPDGTIKRLIYAASTLDSGVPKRFSGSRSGSDYSLT ISSLESEDFADYYCLQYASYPWTFGGGTKLESNGE<br>CDR-L1:QEISGY<br>CDR-L2:AASTLDS CDR-L3:LQYASYPWT |
| 22B3 | MHVQLQESGGGLVKPGGSLKLSCAASGFTFSSYAMSWV RQTPEKRLEWVASISSGGSTYYPDSVKGRFTISRDNARNILY LQMSSLRSEDTAMYFCARRYRYDAWFAYWGQGTTVTVSSK<br>CDR-H1:GFTFSSYA<br>CDR-H2:ISSGGST CDR-H3:ARRYRYDAWFAY | MTQSPAILSASPGEKVTMTCRASSSVSYMHWYQQKPGS SPKPWIYATSNLASGVPARFSGSGSGTSYSLTISRVEAED VATYYCQQWSSNPRGRSVEAPSWKSNG<br>CDR-L1:SSVSY<br>CDR-L2:ATSNLAS<br>CDR-L3:QQWSSNPRGRSV |
| 43C2 | MAAQQSGAELMKPGASVKISCKATGYTFSTYWIEWVKQ RPGHGLEWIGEILPGGGSTNYNEKFKGKAMFTADTSSNT AYMQLSSLASEDSAVYYCARVRYGNYGGNYYAMDYWGQ GTTVTVSS<br>CDR-H1:GYTFSTYW CDR-H2:ILPGGGST CDR-H3:ARVRYGNYG GNYYAMDY | MDIELTQSPASLAVSLGQRATISYRASKSVSTSGYSYMHWNQQK PGQPPRLLIYLVSNLESGVPARFSGSGSGTDFT LNIHPVEEEDAATYYCQHIRELTRSEG<br>CDR-L1:KSVSTSGYSY CDR-L2:LVSNLES<br>CDR-L3:QHIRELTRSE |
| 7E12 | MEVQLQQSGAELVRPGASVKLSCKTSGYIFTSYWIHWVKQRSGQG LEWIARIYPGTGSTYYNEKFKGKATLTADKSSSTAYMQLSS LKSEDSAVYFCARGKYGNLYYFDYWGQGTTVTVSSK<br>CDR-H1:GYIFTSYW<br>CDR-H2:IYPGTGST CDR-H3:ARGKYGNLYYFDY | MDIELTQSPASLAVSLGQRATISYRASKSVSTSGYSYMH WNQQKPGQPPRLLIYLVSNLESGVPARFSGSGSGTDFTL NIHPVEEEDAATYYCQHIRELTRSEGAPSWKSNG<br>CDR-L1:KSVSTSGYSY<br>CDR-L2:LVSNLES<br>CDR-L3:QHIRELTRSEGA |
| 13G11 | MIMVQSQESGPDLVKPSQSLSLTCTVTGYSITSGYSWHWIR QFPGNKLEWMGYIHYSGVTNYNPSLKSRISITRDTSKNQFFLQLNSVT TEDTATYYCARAPLYYGNYVWFSYWGQGTTVTVSSK<br>CDR-H1:GYSITSGYS<br>CDR-H2:IHYSGVT CDR-H3:APLYYGNYVWFSY | MTCDIELTQSPASLSASVGETVTITCRASGNIHNYLAWYQ QKQGKSPQLLVYNAKTLADGVPSRFSGSGSGSQYSLKINSL QPEDFGSYYCQHFWSTPWTFGGGTKLENQTDK<br>CDR-L1:GNIHNY<br>CDR-L2:NAKTLAD<br>CDR-L3:QHFWSTPWT |

([187]INCTYLTL[194]), 18-2 ([195]SSNY[198]), and 18-3 ([199]FYTFFKLY[206]). The six short peptides were identified by i-ELISA and dot-ELISA. In Fig. 6C and D, the error bars refer to standard deviation (SD) of three replicates from one experiment. As shown in Fig. 6C and E, the 14-1 and 14-3 peptides could react with the 18C3 and 7E12 mAbs. As shown in Fig. 6A, the predicted secondary structures of 14-1 and 14-3 had $\beta$-folding and corner structures, hydrophilicity, and high antigenicity, and their surface accessibility coefficients were greater than zero. Although 14-2 also had high antigenicity, its surface accessibility coefficient was less than zero. This indicated that the region is folded inside so that 14-2 could not be recognized by antibodies. Moreover, the 18-1 peptide and the 18-3 peptide did not react with any mAbs (Fig. 6D and F) because of negative antigenicity and surface accessibility coefficients (Fig. 6B), while 18-2 could react with 18A3 and 7E12 mAbs. The sequence of 18-2 contained $\beta$-fold, $\alpha$-helix, and corner structures. In addition, both the antigenicity and surface availability coefficients of 18-2 were positive. Therefore, the B cell epitopes of the extracellular region of the CD2V protein were 14-1 ([147]FVKYT[151]), 14-2 ([157]EYNWN[161]), and 18-3 ([195]SSNY[198]).

## DISCUSSION

ASFV, a large DNA double-stranded virus, is the pathogen of ASF, which is a disease with high pathogenicity and mortality in pigs (3, 29). Since 2007, a devastating ASF outbreak has occurred in the Caucasus, the Russian Federation, the Baltic States, Eastern European countries, and now China (3). ASF poses a serious threat to the pig

**FIG 3** Legend (Continued)

and IFA (D). The 13G11 mAb could react with the CD2V recombinant protein by Western blotting (E) and IFA (F). The 18A3 mAb could react with the CD2V recombinant protein by Western blotting (G) and IFA (H). The 43C2 mAb could react with the CD2V recombinant protein by Western blotting (I) and IFA (J). Positive serum reacted with the CD2V recombinant protein by Western blotting (K) and IFA (L).

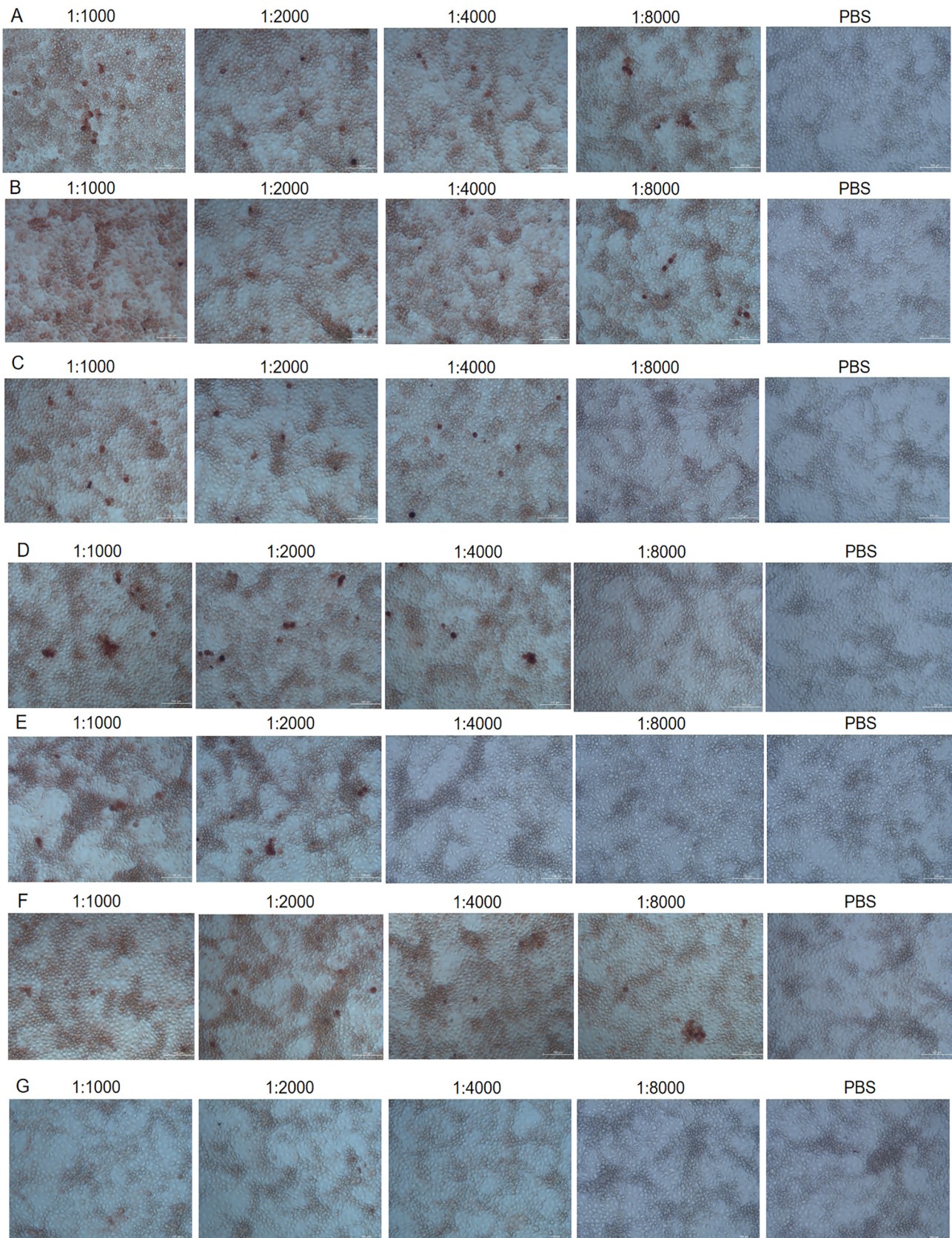

**FIG 4** IPMA immunoperoxidase monolayer assay (IPMA) for CD2V mAbs. CD2V mAb 18A3 (A), 22B3 (B), 43C2 (C), 7E12 (D), and 13G11 (E) as well as ASFV-positive serum (F) could bind specifically to ASFV HLJ/18 strain. The negative control was pig negative serum, the blank control was PBS, and pig negative serum (G) and PBS could not react with ASFV HLJ/18.

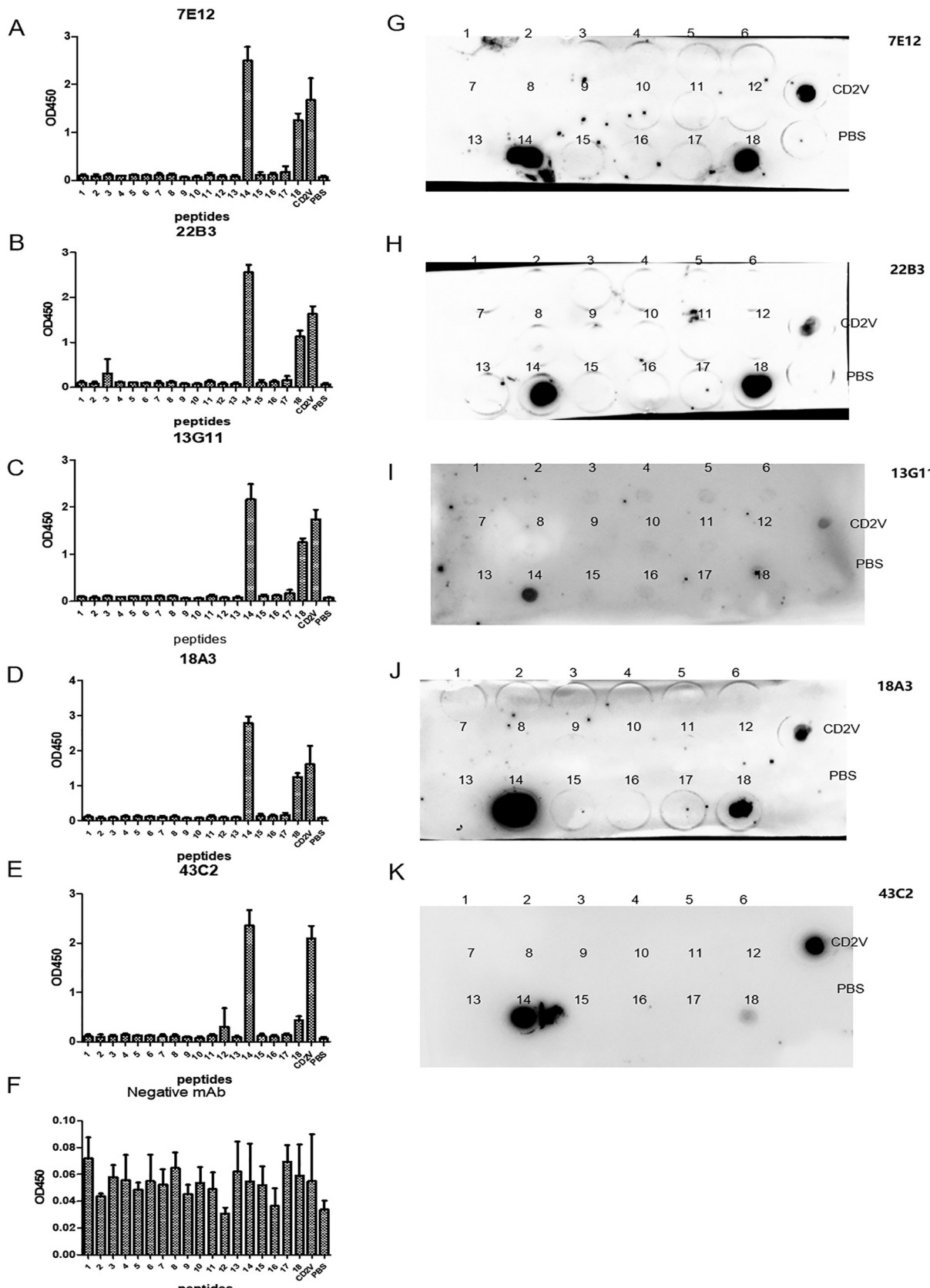

**FIG 5** Preliminary identification of CD2V protein B cell epitopes by the scanning peptide method. Reactions of CD2V mAbs with 18 short peptides were detected by i-ELISA (A to E) and dot-ELISA (G to K). (F) WH303 mAb was the negative control. Each experiment was conducted three times, and the error bars represent standard deviation (SD).

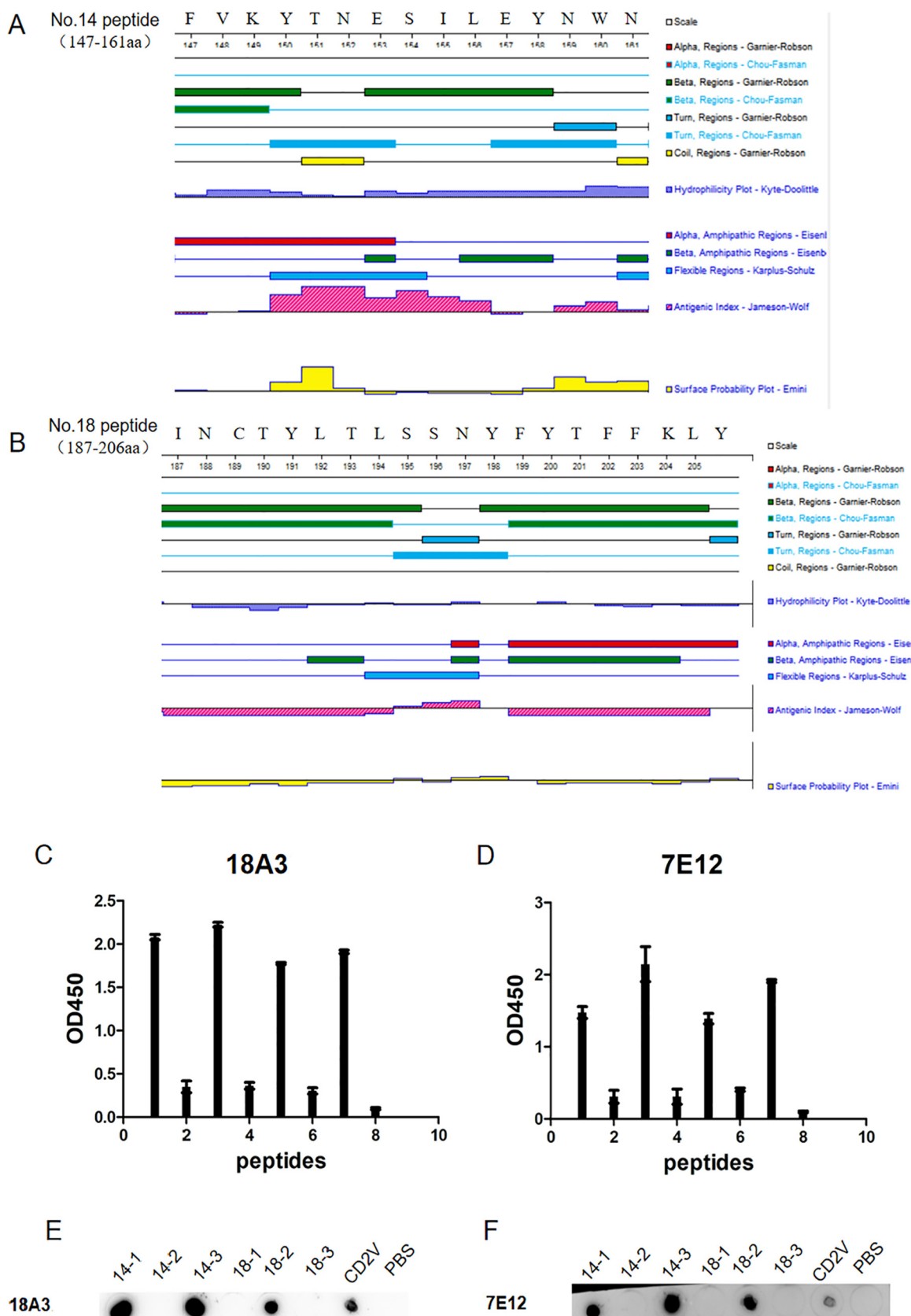

**FIG 6** Truncation of short peptides. (A) Secondary structure analysis of peptide no. 14 by DNASTAR software. (B) Secondary structure analysis of peptide no. 18 by DNASTAR software. Reaction of the mAb 18A3 with two short peptides, no. 14 and no. 18, by i-ELISA (C) and dot-ELISA (E). Reaction of the mAb 7E12 with two short peptides, no. 14 and no. 18, by i-ELISA (D) and dot-ELISA (F). All the experiments were repeated three times and the error bars refer to standard deviation (SD).

industry worldwide (30). There are no commercial vaccines for ASF (31, 32). Due to the lack of knowledge about the protective immunity of ASFV antigens and the understanding about the diversity of these protective antigens in nature, the progress of ASFV vaccine development and disease control has been hindered (2, 33–35).

ASFV p30, p54, p72, CD2V, EP153R, p12, D117L, pp62, and other proteins are the main targets in the current research on ASF vaccines (36, 37). Among them, p72, p54, and p30 proteins, which had neutralizing effects, could inhibit virus to adsorb, and internalize, and activate the cytotoxic T lymphocyte (CTL) response (38). CD2V, p12, and D117L are envelope or inner membrane proteins that also induce the production of neutralizing antibodies and participate in the inhibition of virus invasion and release (39). CD2V protein expressed in baculovirus has been used as an immunogen to immunize pigs that are challenged by the homologous ASFV E75 strain and that show dose-dependent protection. Antibodies induced by CD2V inhibit erythrocyte adsorption, temporarily inhibit infection, and play an additional immunoprotective role (40). In the process of ASFV infection, CD proteins target the regulatory trans-Golgi network (TGN) protein complex AP-1, which is a key element in cellular trafficking, to initiate virus entry (20, 41). CD2V proteins are the main proteins used as targets for ASFV vaccines (20, 41).

The CD2V protein contains a signal region (amino acids 1 to 16), extracellular domain (amino acids 17 to 206), transmembrane region (amino acids 207 to 229), and cytoplasmic domain (amino acids 230 to 360). The extracellular domain of the CD2V protein is the immunodominant antigen during ASFV infections because epitopes exist on the extracellular domain (21). One strategy to increase the immunoreactivity of the CD2V protein is to increase the number of epitopes on the CD2V protein (42, 43). In the current study, two CD2V extracellular regions were linked by a flexible linker (GGGGSGGGGSGGGGS) to form a CD2V recombinant protein. The antibody titer induced by the CD2V recombinant protein was 10 to 100 times higher than that induced by the monomer. The immunogenicity and antigenicity of the CD2V recombinant protein were improved.

Neutralizing antibodies have been proved to play key roles in the protective responses against most viruses (39, 44). Because of the humoral immunity which caused the protective immune response of ASF, antibodies could partially protect pigs from ASFV infection (45). At present, neutralizing antibodies against the ASFV p30, p54, and p72 proteins have been studied (46). Although neutralizing antibodies against the ASFV p30, p54, p72, and p22 proteins could not completely protect pigs, neutralizing antibodies against p30 and p54 can jointly provide partial protection against ASFV infection (47). Thus far, there has been no research on antibodies against the CD2V protein. In this study, mAbs against the CD2V protein were prepared for the first time. The heavy chain and light chain of CD2V-neutralizing mAbs were sequenced for the first time. The CDRs of the heavy chain and light chain of each mAb were labeled. Three B cell epitopes of the CD2V protein were identified by CD2V mAbs: [147]FVKYT[151], [157]EYNWN[161], and [195]SSNY[198]. The results of primary screening for CD2V B cell epitopes indicated that all the mAbs could recognize two distinct linear peptides. The experiment results were in keeping with bioinformatics prediction. Moreover, although most epitopes were not linear, two mAbs could recognize three linear epitopes ([147]FVKYT[151], [157]EYNWN[161], and [195]SSNY[198]). This research promoted basic understanding of the gene location, structure-function relationship, antigen variation, and mechanism of ASFV CD2V protein and supported the development of ASF-preventing live attenuated vaccines (48), synthetic peptide vaccines, recombinant vaccines, anti-idiotypic vaccines, and others.

In short, a CD2V recombinant protein was designed specifically to improve antigenicity and immunogenicity, which was indicated by structural analysis. The CD2V recombinant protein was expressed in the Bac-to-Bac baculovirus expression vector system and purified by Ni-affinity chromatography. Five mAbs of CD2V were developed, and B cell epitopes of CD2V were screened out. This study provides theoretical support for the development of subunit vaccines and diagnostic reagents.

## MATERIALS AND METHODS

**Genes, cells, and animals.** ASFV China/2018/AnhuiXCGQ isolated strain (GenBank accession number MK128995.1) was used in this study. ASFV EP402R gene encoding CD2V protein was synthesized in Shanghai Sangon Bioengineering Co., Ltd. (Shanghai, China). Sf21 cells and hybridoma cells (SP2/0) were stored in Key Laboratory of Animal Immunology of Henan Academy of Agricultural Sciences. Ten healthy female BALB/c mice, which were about 20 g and 6 to 8 weeks old, were provided by Key Laboratory of Animal Immunology of Henan Academy of Agricultural Sciences.

**Preparation of ASFV CD2V recombinant protein.** Recombinant shuttle Bacmid for CD2V recombinant protein was selected by blue-white spot screening and transfected into Sf21 cells to make baculovirus for CD2V recombinant protein. Expression of CD2V was identified by SDS-PAGE, Western blotting, dot-ELISA, and indirect immunofluorescence assay (IFA). CD2V protein with different does, 0.5 $\mu$g/mL, 1.0 $\mu$g/mL, and 1.5 $\mu$g/mL, was dotted on nitrocellulose filter membrane (NC) in dot-ELISA. ASFV CD2V recombinant protein was produced after virus amplification for three generations. Cell supernatants were then collected by centrifugation for 10 min at 3,000 $\times$ $g$, and soluble proteins were purified through a 5 mL HisTrap EXCEL column (GE Healthcare). To be specific, impurity protein was washed by washing buffer (0.01 M PBS with 10 mM iminazole) and target protein CD2V was eluted by elution buffer (0.01 M PBS with 100 mM iminazole). CD2V recombinant protein was analyzed by SDS-PAGE and Western blotting to verify protein size and purity. Specifically, mouse anti-6×His tag monoclonal antibody (Proteintech, Wuhan, China) was used as the primary antibody at 37°C for 1 h, and goat anti-mouse antibody labeled with horseradish peroxidase (HRP; Abcam, England) was used as the secondary antibody at 37°C for 1 h in Western blotting and dot-ELISA. Mouse anti-6×His tag monoclonal antibody (Proteintech, Wuhan, China) was used as the primary antibody at 37°C for 1 h, and goat polyclonal antibody (pAb) to mouse IgG labeled with Alexa Fluor 488 (Abcam, England) was used as the secondary antibody at 37°C for 1 h in IFA.

The reactivity of CD2V recombinant protein with ASFV-positive serum was detected by i-ELISA, which used mouse immune serum from 1:50 to 1:26,500 double ratio diluted as a primary antibody and the HRP-labeled goat anti-mouse IgG diluted to 1:1,000 as a secondary antibody.

The purified CD2V protein was diluted to 0.1 mg/mL 8 $\mu$L, dropped to the 100-mesh ordinary carbon support film (Mirror Technology Co., Ltd., Beijing, China), and incubated for 2 min at room temperature (RT). Then, the protein was carefully removed from the film. Phosphotungstic acid was dropped to the center of the film, incubated at RT for 2 min, and dried overnight. The samples were sent to Frontier Research Facility of cryo-electron microscopy (cryo-EM) in the Center of Advanced Analysis and Gene Sequencing in Zhengzhou University to observe the results by 120 kV TEM (Thermo Fisher, USA).

Meanwhile, the molecular weight of CD2V protein was identified by size exclusion chromatography (HiLoad Superdex 200 pg, cat no. 28989335, Cytiva, USA). The steps were as follows. First, open the computer and access the AKTA system. Then, turn on AKTA power supply, connect the system, and enter UniCorn system. After washing pump A with water, set the upper pressure limit and the appropriate flow rate of the chromatography column. Connect the HiLoad Superdex 200 pg chromatography column to the fast-performance liquid chromatographer (FPLC). Flush the chromatography column with at least 1 volume of water. Balance the chromatography column with at least 1 column volume of buffer (0.01 mM PBS). After centrifuging the prepared sample at 12,000 rpm at 4°C for 10 min, load the sample through loading sample ring in load state. The sample enters the system under inject state and returns to load state at 2 mL.

The particle size of CD2V protein was detected by ZetaCAD (Malvern's Zetasizer, England). CD2V protein was diluted to 1 mg/mL by pure water in cuvette. Then, we put the cuvette into the ZetaCAD and detected it.

**Preparation and selection of monoclonal antibodies to CD2V recombinant protein.** The BAlB/c mouse with the highest serum titer was selected to hyperimmune with 10 ng of ASFV CD2V protein which was diluted to 500 $\mu$L by 0.01 M PBS. Three days later, spleen cells of hyperimmune mouse were fused with myeloma cell Sp2/0. Hybridoma cells were cultivated at 37°C 5% CO$_2$. Ten days later, positive hybridoma cells were selected by indirect ELISA. A total of 1 $\mu$g/mL purified CD2V protein was packaged into the enzyme-linked immunoreactive well at 100 $\mu$L per well at 4°C overnight. After washing with PBST (0.01 M PBS, 0.05% Tween 20) 3 times, 300 $\mu$L 5% skim milk was added to per well at 37°C for 2 h. One hundred microliters of cellular supernatant was added to the per reaction well as the primary antibody. Meanwhile, a positive control (serum sampled from hyperimmune mouse eyes), a negative control (serum sampled from mouse injected with 0.01 M PBS), and a blank control (0.01 M PBS) were set. Primary antibody was incubated at 37°Cfor 30 min. After washing 3 times with PBST, goat anti-mouse antibody labeled with horseradish peroxidase (Abcam, England) was diluted to 1:1,000 as secondary antibody and incubated at 37°C for 30 min. Then, the reacting results were colored with tetramethylbenzidine [(TMB); NCM Biotech, Jiangsu, China]. Positive monoclonal antibodies (mAbs) to CD2V recombinant were prepared by subcloning.

**Characterization of mAbs to CD2V recombinant protein.** Characterization of mAbs to CD2V recombinant protein was analyzed. Mouse Ig subtype of monoclonal antibody was detected by using mouse monoclonal antibody isotyping ELISA kit (Proteintech, cat. no. KMI-2, Wuhan, China). The variable region of heavy chain and light chain of the monoclonal antibody was amplified by PCR. The PCR used the primer sequences in Table S2. The amplified products were recovered after gel cut using an omega nucleic acid gel recovery kit and were sequenced by Shanghai Sangon Biotechnology Co., Ltd. to obtain the heavy-chain (VH) and light-chain (VL) regions of mAbs. The CDRs in both the VH and VL for the five CD2V recombinant mAbs were highly variable. Using the international ImMunoGeneTics information system (IMGT) (http://www.imgt.org/3Dstructure-DB/cgi/DomainGapAlign.cgi) to verify, the sequences

of three heavy-chain variable regions were labeled CDR-H1, CDR-H2, and CDR-H3, and the sequences of three light-chain variable regions were labeled CDR-L1, CDR-L2, and CDR-L3.

**Titration of mAb staining of infected monolayers by IPMA.** The inactivated plate, which was inoculated with porcine alveolar macrophages (PAMs) and infected with ASFV HLJ/18 strain, was provided by Harbin Veterinary Research Institute and stored at −20°C. The plate was taken out from −20°C, blocked again with 5% skim milk at 37°C for 1 h, and washed with PBST 3 times. Monoclonal antibody ascites, ASFV-positive serum, and ASFV-negative serum of ASFV-recovered pigs were diluted into 1:1,000, 1:2,000, 1:4,000, and 1:8,000 as primary antibodies. At the same time, PBS was used as blank control. A total of 100 µL of primary antibodies was added into the enzyme-labeled well of the plate, incubated at 37°C for 30 min, and then washed with PBST 3 times. A total of 100 µL of HRP-conjugated goat anti-mouse IgG diluted 1:1,000 was added into the well, incubated at 37°C for 30 min, and then washed with PBST for 3 times. A total of 100 µL of 3-amino-9-ethylcazole (AEC) dye was added to each well, stained for 10 min at room temperature, and observed under an inverted microscope. The positive well was brownish red, while the blank control well was colorless. Under the microscope, the cells in the positive well were stained brownish red while cells in the blank control well was not stained. The negative pores were colorless and the cells were not stained under microscope.

**Identification of ASFV CD2V protein B cell epitopes.** The extracellular region of ASFV CD2V protein was overlapped and truncated into 18 short peptides by peptide scanning method, i.e., overlapping polypeptide method. Among them, each sequence of no.1 to 17 short peptide was 15 amino acids (aa) except no. 18 short peptide with 21 aa. Five amino acids were overlapped between the segments (Fig. 7), and all polypeptides were synthesized by Gill Bio (Shanghai, China). The short peptide was conjugated with IgG-free bovine serum albumin (BSA; Thermo Fisher, US) by carbodiimide (EDPC) method (49). Positive short peptides which can react with mAbs were selected by indirect ELISA and dot-ELISA with 2.5 µg/mL of each short peptide and CD2V recombinant protein as coating antigens. In the experimental group, 18A3, 22B3, 43C2, 7E12, and 13G11 mAbs diluted to 1:1,000 (about 30 µg/mL) were used as primary antibodies, and HRP-labeled goat anti-mouse IgG was used as the secondary antibody. In the control group, WH303 monoclonal antibody to pestiviruses (Classical Swine Fever Specific) diluted 1:1,000 and PBS were used as negative control and blank control, respectively. Then, compared with the possible epitope regions analyzed by bioinformatics software, positive peptides were truncated into shorter segments to verify the position of the functional B cell epitope of CD2V protein. The shorter peptides which were conjugated with IgG-free BSA were used as coating antigens in indirect ELISA to identify their reaction with monoclonal antibody. The indirect ELISA accurately determined the position of CD2V protein B cell epitope and displayed in the tertiary structure of CD2V protein by homologous modeling. For indirect ELISA, all results were repeated three times.

**Ethic Statement.** All animal treatments were carried out in accordance with the National Institutes of Health Guide for the Care and Use of Laboratory Animals, and approved by the Institutional Animal

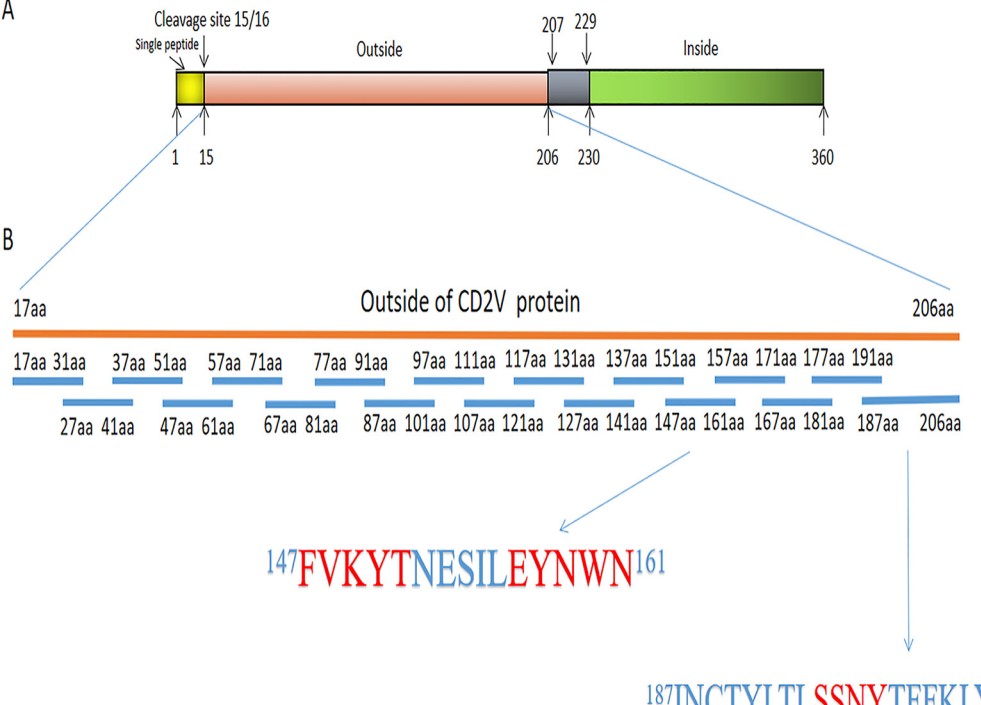

**FIG 7** Short peptides designed by overlapping polypeptide method to the extracellular region of CD2V. (A) Analysis of full-length sequence of CD2V protein. (B) Eighteen short peptides designed to the extracellular region of CD2V and sequences of no. 14 peptide and no. 18 peptide.

Care and Use Committee of the Key Laboratory of Animal Immunization of Henan Academy of Agricultural Sciences.

**Data availability.** All data presented in this study are available on request from the corresponding authors.

## SUPPLEMENTAL MATERIAL

Supplemental material is available online only.

**SUPPLEMENTAL FILE 1**, PDF file, 1.3 MB.

## ACKNOWLEDGMENTS

We thank the Key Laboratory of Animal Immunization of Henan Academy of Agriculture Sciences for the Lab support.

The study was supported by Zhongyuan high level talents special support plan (20420051005), National Natural Science Foundation of African Swine Fever Key Project (31941003), and National Natural Science Foundation project (31941001).

Aiping Wang and Gaiping Zhang are the correspondence authors; who plays a guiding role in study design, laboratory supervision, and manuscript editing. Rui Jia contributed to study design, do experiments, data analysis, and manuscript drafting, editing, and writing. Yilin Bai contributed to develop the monoclonal antibodies. Yilin Bai and Hongliang Liu contributed significantly to collection of laboratory data. Yumei Chen, Jingming Zhou, and Hua Feng contributed to laboratory quality control and data collection. Yilin Bai polished the language of the manuscript. Peiyang Ding, Yuanyuan Tian and Mingyang Li helped perform the analysis with constructive discussions. Gaiping Zhang and Aiping Wang contributed to study design, laboratory supervision, and manuscript editing.

We declare no conflict of interest.

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
