## [Reviewer comments · Microbiology Spectrum]

Microbiology Spectrum

Identification of linear B cell epitopes on CD2V protein of African swine fever virus by monoclonal antibodies

Rui Jia, Gaiping Zhang, Hongliang Liu, Yumei Chen, Peiyang Ding, Jingming Zhou, Hua Feng, Mingyang Li, Yuanyuan Tian, and Aiping Wang

Corresponding Author(s): Aiping Wang, Department of Bioengineering , Zheng zhou University , Zheng zhou

Review Timeline:

Submission Date:	July 26, 2021
Editorial Decision:	August 24, 2021
Revision Received:	October 27, 2021
Editorial Decision:	December 6, 2021
Revision Received:	January 21, 2022
Accepted:	January 24, 2022

Editor: Fikri Avci

Reviewer(s): The reviewers have opted to remain anonymous.

Transaction Report:

DOI: <https://doi.org/10.1128/Spectrum.01052-21>

August 24, 2021

Dr. Gaiping Zhang
Sichuan Agricultural University&Henan Provincial Key Laboratory of Animal Immunology, Henan Academy of Agricultural Sciences
No.116 Huayuan Road
Zheng zhou
China

Re: Spectrum01052-21 (Development and application of neutralizing monoclonal antibodies against a modified African swine fever virus CD2V protein)

Dear Dr. Gaiping Zhang:

Thank you for submitting your manuscript to Microbiology Spectrum. Two experts reviewed your manuscript and while they found the findings valuable to the field, they had concerns that need to be addressed in a revised manuscript experimentally or textually as indicated in the reviewers' comments.

When submitting the revised version of your paper, please provide (1) point-by-point responses to the issues raised by the reviewers as file type "Response to Reviewers," not in your cover letter, and (2) a PDF file that indicates the changes from the original submission (by highlighting or underlining the changes) as file type "Marked Up Manuscript - For Review Only". Please use this link to submit your revised manuscript - we strongly recommend that you submit your paper within the next 60 days or reach out to me. Detailed information on submitting your revised paper are below.

Link Not Available

Sincerely,

Fikri Avci

Journals Department
Reviewer comments:

Reviewer #1 (Comments for the Author):

In this manuscript, Jia and colleagues describe the isolation and characterization of five mAbs that bind to the CD2V protein of African Swine Fever Virus (ASFV). The authors measure neutralization, reactivity by Western blot, and reactivity by ELISA and use molecular docking to hypothesize epitopes that may be targeted by each antibody. The authors also generate single-chain variable fragments (scFvs). The description of mAbs is valuable to the field, but there are some issues with presentation and conclusions that, if improved, would greatly strengthen the manuscript.

Major points:

1) There is probably an opportunity to condense several figures together for the sake of clarity. For example. Figures 1 and 2 describing expression and purification of CD2V could be combined. While inclusion of sequences is greatly appreciated (Figure

5), this information would probably be better suited as a table

2) The neutralization data presented appears convincing in that there is a significant reduction in fluorescence signal. However, this does not give much information as to the potency of the mAbs since the authors only test a single concentration (and this concentration is not defined in the methods). It is important for the authors to state the concentration (ug/mL) of mAb used in the neutralization experiments at a minimum. Carrying out the neutralization assay as a dose-response would give valuable information as to the potency of the mAbs.

3) While the ELISA signal observed for mAb binding to the peptides is robust, I am somewhat concerned in that this experiment was also conducted at a single concentration, and there is not a negative control mAb used in the ELISA. The inclusion of a negative control mAb is especially important given that all the mAbs seem to have essentially the same reactivity pattern. This experiment should be repeated with the appropriate controls and preferably as a dose-response assay. At a minimum, the concentration of mAb should be provided, which is not mentioned in the ELISA methods.

4) I find it surprising that the ELISA data indicates that all the mAbs recognize two distinct linear peptides - the authors should comment here on whether that aligns with their expectations. Given that most neutralizing epitopes are not linear, the authors should also comment on the likelihood that all five mAbs recognize linear epitopes and neutralize.

5) The rationale for scFv generation was unclear, the authors should discuss.

6) I am unconvinced by the docking experiments and the authors' conclusion that they have identified critical contact residues, as these residues are not experimentally validated. In all the simulated structures of the scFvs used for docking, the heavy and light chain appear to be splayed apart, which would be highly unexpected. The contacts are also extremely variable across different scFv simulations and some appear to not take place at the expected paratope interface (where the CDRs are located and where the vast majority of Ab:antigen interactions occur). Due to these issues, these data appear to not reflect the biological reality and should be omitted from the manuscript.

7) Line 79: it would be good to insert references for each function of CD2V here to facilitate the ability of the reader to look up prior literature.

8) More detail as to hybridoma generation (fusion cell line, procedure, etc) should be included.

Other points:

1) It would be good for the authors to put the names of each mAb on the panels in Fig 3 rather than having them only in the legend

2) Fig 1 legend ends mid-sentence

3) Fig 7: would be more informative to put what the positive control is on the figure rather than just saying "PC"

4) Fig 4 could be moved to supplemental

5) There are a few grammatical or idiomatic issues - e.g. in neutralization assay methods "poisoned" should be replaced with "infected" or "inoculated". Similarly, "deletion" should be replaced with "dilution" in the same section.

Reviewer #2 (Comments for the Author):

The manuscript describes the generation of a CD2-like protein construct from African swine fever virus, and the generation of mouse monoclonal antibodies, as well as some immunological characterization of these antibodies.

Several of the supplemental figure references do not match those in the supplemental material.

Much of the results contains methodology that should be moved to the methods section

Line 130: Data needs to be shown for the mAb binding.

Line 139: clarify what is meant by best reactivity

The negative-stain EM image in Figure 2D is poor quality, and perhaps is not stained well. This needs to be redone and please explain why nanoparticles are expected. Please also analyze the protein by size exclusion chromatography to identify if the nanoparticles are actually formed in solution.

Check Figure 5, the highlighted regions for the CDRs are missing a couple residues in some cases.

The B cell epitope prediction nicely matches experimental results with peptides.

Figure 4 should be a supplemental figure.

Figure 8 and 9 have error bars but there is no indication of what those represent or how many replicated were conducted

The neutralization assays are poorly conducted. No methods are given for this assay, and a more refined dilution curve is needed rather than just showing one image with a single dilution. The positive control and negative control identities need to be provided. A negative control should be a mouse isotype control, and it would be good to include positive and negative mouse immunization serum for good measure.

The docking experiment not scientifically sound. Typically, in docking one would have structures of one or both of the pieces, and in this case, docking is attempted with predicted models of both pieces. In particular, antibody CDR3 loops are difficult to model, while also being the most important in binding. All of these sections should be removed from the manuscript. The antibody structure prediction does not add much to the manuscript. Additionally, the heavy and light chains are separated in all the simulations, which very likely not the case. The docking prediction of CD2V is okay since some is verified with B cell epitope data.

Methods needed for ELISAs, dot blots, neutralization assays, western blots, and microscopy.

Some typos throughout

Staff Comments:

Preparing Revision Guidelines

Please return the manuscript within 60 days; if you cannot complete the modification within this time period, please contact me. If you do not wish to modify the manuscript and prefer to submit it to another journal, please notify me of your decision immediately so that the manuscript may be formally withdrawn from consideration by Microbiology Spectrum.

If you would like to submit an image for consideration as the Featured Image for an issue, please contact Spectrum staff.

Dear reviewers:

Thank you for your comments. We were in response to your comments carefully.

For Reviewer 1:

Major points:

1. We combined Figures 1 and 2 as expression and purification of CD2V. And we revised data of Figure 5 to a table named Table 3.
2. The neutralization experiment was completed in biosafety protection Level 3 laboratory of Institute of Military Veterinary Medicine, Academy of Military Medical Sciences (Jilin, Changchun). However, we could not added the dose-response experiments of neutralization assay influenced by COVID-19 because we could not go to there. Before neutralization assay, we carried on IPMA assay, and found that the IPMA titers of ASFV mAbs 18A3, 22B3, 43C2, 7E12 and 13G11 were 1:8,000, 1:8,000, 1:4,000, 1:4,000, and 1:2000, respectively. And we carried out the neutralization experiments with the concentration of 18A3 and 22B3 was 3.75 μ g/mL(1:8000), the concentration of 43C2 and 7E12 was 7.5 μ g/mL(1:4000), the concentration of 13G11 was 15 μ g/mL(1:2000). Please understand that situation.
3. We provided the concentration of mAb was about 30 μ g/ml (diluted to 1:1000) in the indirect-ELISA method. And we used WH303 monoclonal antibody to pestiviruses (Classical Swine Fever Specific) (Veterinary Laboratories Agency, England) as the negative control mAbs. The results showed that WH303 mAbs could not react to eighteen short peptides in that OD450 values were lower than 0.5.
4. We added the comments about "whether that aligns with their expectations", that is, "The results of primary screening for CD2V B cell epitopes indicates that all the mAbs recognized two distinct linear peptides, which aligned with bioinformatics prediction. ". And we commented on "the likelihood that all five mAbs recognize linear epitopes and neutralize", that is, "Moreover, although most neutralizing epitopes are not linear, two neutralizing mAbs could recognize three linear epitopes (147FVKYT151, 157EYNWN161 and 195SSNY198)".
5. In this study, the ScFv generation is for molecular docking of antibody and antigen. We deleted the docking section according to your suggestion 6. So we also deleted the ScFv generation and did not discuss the rationale for ScFv generation.
6. We deleted the docking section according to your suggestion.
7. Line 79 changed to Line 64-65: we inserted references for each function of CD2V. That is as follows:
 20. Daniel PN, *et al.* (2015), CD2v Interacts with Adaptor Protein AP-1 during African Swine Fever Infection, *PLoS One*, **10**, e123714.
 21. Dixon LK, *et al.* (2004), African swine fever virus proteins involved in evading host defence systems, *Veterinary Immunology & Immunopathology*, **100**, 117-134.
 22. Tulman ER, Delhon GA, Ku BK, Rock DL (2009), African Swine Fever Virus, *Current Topics in Microbiology and Immunology*, **328**, 43-87.
8. We added details of hybridoma generation to the Genes, cells and animals of METHODS AND MATERIALS.

Other points:

1. We put the names of each mAb on the panels in Fig 3.

2. We revised the legend of Fig.1.
3. Fig 7 changed to Fig 5. We marked that the positive control is ASFV positive serum.
4. We moved Fig 4 to supplemental materials as Figure S5.
5. We revised a few grammatical or idiomatic issues, for instance, "poisoned" was revised as "infected" and "deletion" was revised as "dilution" in neutralization assay methods.

Reviewer 2

1. We checked the supplemental material carefully to ensure the figures references match.
2. We checked the results carefully and removed the methodology to the methods section.
3. Line 130 changed to Line 111-118, we provided the data (OD450 value) of isotypes identification in Figure 2 and manuscript.
4. Line 139 changed to Line 124-128. We revised "the best reactivity" to "the better reactivity". That is, Comparison of the reaction between positive serum of rehabilitated pigs infected with ASFV and CD2V protein (Fig. 3K), the bands of Western blots for CD2V recombinant protein with the monoclonal antibody was wider than that for CD2V recombinant protein with ASFV positive serum. It showed that CD2V recombinant protein had the better reactivity with the monoclonal antibody than positive serum.
5. Figure 2D changed to Figure 1G. This figure was redone by sending to Frontier Research Facility of Cryo-EM in Center of Advanced Analysis and Gene Sequencing in Zhengzhou University to observe the results by 120kv TEM (Thermo Fisher, US). We expected the CD2V recombination protein was nanoparticles, because recombination protein, which gathered into virus-like particles (VLP), could have strong immunogenicity and biological activity. Exclusion chromatography identified that the size of protein particles was about 10nm.
6. We checked Figure 5, and found the CDR-L3 of 22E3 mAbs missing a residue "R".
7. Thank you for your support about the B cell epitope prediction nicely matches experimental results with peptides.
8. We removed the Figure 4 to the supplemental material as a supplemental figure.
9. Figure 8 and 9 changed to Figure 6 and 7. We stated that error bars represented Standard Deviation (SD). All experiments were repeated three times.
10. The neutralization experiment was completed in biosafety protection Level 3 laboratory of Institute of Military Veterinary Medicine, Academy of Military Medical Sciences (Jilin, Changchun). However, we could not add the dose-response experiments of neutralization assay influenced by COVID-19 because we could not go to there. Before neutralization assay, we carried on IPMA assay, and found that the IPMA titers of ASFV mAbs 18A3, 22B3, 43C2, 7E12 and 13G11 were 1:8,000, 1:8,000, 1:4,000, 1:4,000, and 1:2000, respectively. And we carried out the neutralization experiments with the concentration of 18A3 and 22B3 was 3.75µg/mL(1:8000), the concentration of 43C2 and 7E12 was 7.5µg/mL(1:4000), the concentration of 13G11 was 15µg/mL(1:2000). Please understand that situation. Thank you for your understanding.
11. We deleted docking experiment according to your comments.
12. Methods for ELISAs, dot blots, neutralization assays, western blots, and microscopy were described in detail.
13. We revised some typos throughout.

Thank you for your advice. And I am looking for your response as soon as possible.

Sincerely,
Gaiping Zhang
Zhengzhou University
Zhengzhou 450000, PR China
Email: zhanggaip@126.com

December 6, 2021

Dr. Aiping Wang
Department of Bioengineering , Zheng zhou University , Zheng zhou
zhengzhou
China

Re: Spectrum01052-21R1 (Development and application of neutralizing monoclonal antibodies against a modified African swine fever virus CD2V protein)

Dear Dr. Aiping Wang:

Thank you for the privilege of reviewing your work. When submitting the revised version of your paper, please provide (1) point-by-point responses to the issues raised by the reviewers as file type "Response to Reviewers," not in your cover letter, and (2) a PDF file that indicates the changes from the original submission (by highlighting or underlining the changes) as file type "Marked Up Manuscript - For Review Only". Please use this link to submit your revised manuscript - we strongly recommend that you submit your paper within the next 60 days or reach out to me. Detailed instructions on submitting your revised paper are below.

Link Not Available

Sincerely,

Fikri Avci

Journals Department
Reviewer comments:

Reviewer #1 (Comments for the Author):

In a revised manuscript, Jia and colleagues include some new data and have revised the structure of the manuscript to condense several figures. The figure changes have improved the manuscript and are appreciated. However, there are some additional points that the authors should clarify, with respect to interpretation of their findings.

Major points:

1) In new data in Figure 4, the authors present IPMA "neutralization assay" results. However, this appears to not be a neutralization assay, but rather a measure of the ability of the mAbs to stain infected cell monolayers. While this staining is a nice confirmation of the ability of the mAbs to bind to ASFV CD2V, it does not give information as to whether the antibodies studied prevent viral replication and restrict spread. As a result, the name of this assay should be corrected, possibly to something like "Titration of mAb staining of infected monolayers by IPMA"

2) To that point, the potency of the mAbs in neutralization assays remains unknown. While the authors now provide some measure of the mAb concentration that was used in neutralization assays, the fact the neutralization assay was done at a single concentration gives little information as to the potency of the mAbs. Though the authors note that due to BSL-3 availability they were unable to complete dose-response experiments, I would recommend the authors revise the abstract to remove the claim that "This study provides the most efficient neutralizing epitopes," as the potency of the mAbs remains unknown. If the mAb

IC50s are in the $\mu\text{g}/\text{mL}$ range, they are unlikely to play a physiological role in ASFV neutralization, as the concentrations in serum for them to neutralize would be prohibitively high.

Minor points:

- 1) In Figure 3, "recombination protein" is probably more appropriate as "recombinant protein"
- 2) There are a couple of other instances of grammar or word use where the editorial process might need more attention

Reviewer #2 (Comments for the Author):

Line 114: It indicated that CD2V recombination protein expressed in Bac-to-Bac baculovirus expression vector system had strong immunogenicity and biological activity.
There is no data to support the statement.

Line 110: Please explain why the protein forms nanoparticles. Is this expected?

Figure 1 caption: Indicate that negative-stain EM was used. There is also no reference to the size exclusion in 1H.

There are no methods for the size exclusion chromatography, no information on what type of column was used, and no standards to suggest the 10 nm size is in anyway correct.

Use recombinant protein instead of recombination protein throughout the manuscript and figures.

The CDR3 regions in Table 3 are still incorrect. The C and W define the boundaries for HCDR3. Please verify CDRs in IMGT and provide these analyses to the reviewer. I would do, but I don't have the nucleotide sequences.

Lines 145-162 are really not necessary, just refer to Table 3.

Figure 4, is the negative control pig negative serum or PBS or a mixture? It is not clear from the figure caption.

In Figure 6 and 7, please clarify. Does "the experiment was repeated three times" mean that the bars are the average of those three experiments or three replicates from one experiment?

Table 2 should be moved to the supplemental material.

The manuscript still needs some work on the flow of the presentation.

Staff Comments:

Preparing Revision Guidelines

Please return the manuscript within 60 days; if you cannot complete the modification within this time period, please contact me. If you do not wish to modify the manuscript and prefer to submit it to another journal, please notify me of your decision immediately so that the manuscript may be formally withdrawn from consideration by Microbiology Spectrum.

Reviewer #1 (Comments for the Author):

Thank you so much for your comments.

Major points:

1) In new data in Figure 4, the authors present IPMA "neutralization assay" results. However, this appears to not be a neutralization assay, but rather a measure of the ability of the mAbs to stain infected cell monolayers. While this staining is a nice confirmation of the ability of the mAbs to bind to ASFV CD2V, it does not give information as to whether the antibodies studied prevent viral replication and restrict spread. As a result, the name of this assay should be corrected, possibly to something like "Titration of mAb staining of infected monolayers by IPMA"

Respond to reviewer: Thank you for your suggestion. We corrected the name of the assay to "Titration of mAb staining of infected monolayers by IPMA".

2) To that point, the potency of the mAbs in neutralization assays remains unknown. While the authors now provide some measure of the mAb concentration that was used in neutralization assays, the fact the neutralization assay was done at a single concentration gives little information as to the potency of the mAbs. Though the authors note that due to BSL-3 availability they were unable to complete dose-response experiments, I would recommend the authors revise the abstract to remove the claim that "This study provides the most efficient neutralizing epitopes," as the potency of the mAbs remains unknown. If the mAb IC50s are in the $\mu\text{g/mL}$ range, they are unlikely to play a physiological role in ASFV neutralization, as the concentrations in serum for them to neutralize would be prohibitively high.

Respond to reviewer: Thank you for your review. Because of the COVID-19, the Containment level three" laboratories in Beijing was not opened for us. Hence, the dose-response experiments of neutralization assay could not be completed. We removed the claim that "This study provides the most efficient neutralizing epitopes," and the content about neutralizing mAbs and epitopes at your suggestion. And we revised the title to "Identification of linear B cell epitopes on CD2V protein of African swine fever virus by monoclonal antibodies".

Minor points:

1) In Figure 3, "recombination protein" is probably more appropriate as "recombinant protein"

Respond to reviewer: Thank you. We revised "recombination protein" to "recombinant protein" throughout the manuscript and figures.

2) There are a couple of other instances of grammar or word use where the editorial process might need more attention

Respond to reviewer: Thank you for your comment. We corrected word and grammar errors carefully.

Reviewer #2 (Comments for the Author):

Line 114: It indicated that CD2V recombination protein expressed in Bac-to-Bac baculovirus expression vector system had strong immunogenicity and biological activity.

There is no data to support the statement.

Respond to reviewer: Thank you. To be accuracy, we changed "biological activity" to "immunoreactivity". We added the indirect-ELISA method using the ASFV positive serum as primary antibody to prove the high immunoreactivity of

CD2V protein. And the animal immunization was shown the strong immunogenicity of CD2V protein.

Line 110: Please explain why the protein forms nanoparticles. Is this expected?

Respond to reviewer: Thank you for your comment. The protein forms nanoparticles because of galactosylated modification. This is expected the CD2V to form nanoparticles by posttranslational modification, because CD2V by posttranslational modification could be close to the natural conformation.

Figure 1 caption: Indicate that negative-stain EM was used. There is also no reference to the size exclusion in 1H.

There are no methods for the size exclusion chromatography, no information on what type of column was used, and no standards to suggest the 10 nm size is in anyway correct.

Respond to reviewer: Thank you for your comment. Size exclusion chromatography is a standard chromatographic technique that allows the separation of molecules or molecule complexes by their hydrodynamic volume (grossly equivalent to the molecular weight). We added detailed steps of the size exclusion chromatography and information on what type of column. Therefore, we could estimate the molecular weight according to the relation between the molecular weight of standard sample and the volume of eluent. We used Zeta CAD to verify the particle size of CD2V in solution.

Use recombinant protein instead of recombination protein throughout the manuscript and figures.

Respond to reviewer: Thank you for your comment. We used recombinant protein instead of recombination protein throughout the manuscript and figures.

The CDR3 regions in Table 3 are still incorrect. The C and W define the boundaries for HCDR3. Please verify CDRs in IMGT and provide these analyses to the reviewer. I would do, but I don't have the nucleotide sequences.

Respond to reviewer: We used IMGT to verify the CDRs and provided these analyses in supplemental material Figure S6.

Lines 145-162 are really not necessary, just refer to Table 3.

Respond to reviewer: We removed lines 145-162 according to your suggestion. Thank you.

Figure 4, is the negative control pig negative serum or PBS or a mixture? It is not clear from the figure caption.

Respond to reviewer: We marked clearly that the negative control was pig negative serum and the blank control was PBS in the manuscript and the figure caption.

In Figure 6 and 7, please clarify. Does "the experiment was repeated three times" mean that the bars are the average of those three experiments or three replicates from one experiment?

Respond to reviewer: We clarified the bars are the Standard Deviation (SD) of those three repeat experiments.

Table 2 should be moved to the supplemental material.

Respond to reviewer: We moved the Table 2 to the supplemental material as Table 2S.

The manuscript still needs some work on the flow of the presentation.

Respond to reviewer: We carefully revised the words and grammar of the presentation. Thank you for your patience and kindness.

January 24, 2022

Dr. Aiping Wang
Department of Bioengineering , Zheng zhou University , Zheng zhou
zhengzhou
China

Re: Spectrum01052-21R2 (Identification of linear B cell epitopes on CD2V protein of African swine fever virus by monoclonal antibodies)

Dear Dr. Aiping Wang:

Your manuscript has been accepted, and I am forwarding it to the ASM Journals Department for publication. You will be notified when your proofs are ready to be viewed.

Sincerely,

Fikri Avci
Editor, Microbiology Spectrum
